# The Effects of ESC/ESH-Based Written Generic Lifestyle Advice and a Nutraceutical on 24-Hour Blood Pressure in Patients with High–Normal Office Blood Pressure and Low–Moderate Cardiovascular Risk

**DOI:** 10.3390/nu15245099

**Published:** 2023-12-13

**Authors:** Matteo Landolfo, Francesco Spannella, Chiara Poliseno, Adriano Massacesi, Federico Giulietti, Roberto Festa, Enrico Cavazzin, Giulio Sasso, Alberto Mazza, Riccardo Sarzani

**Affiliations:** 1Internal Medicine and Geriatrics, ESH Excellence Hypertension Centre, IRCCS INRCA, 60127 Ancona, Italy; m.landolfo@inrca.it (M.L.); r.sarzani@univpm.it (R.S.); 2Department of Clinical and Molecular Sciences, Centre for Obesity, University “Politecnica delle Marche”, 60126 Ancona, Italy; 3Area Vasta 3, 62100 Macerata, Italy; 4ESH Excellence Hypertension Unit, Department of Internal Medicine, Rovigo General Hospital, 45100 Rovigo, Italyalberto.mazza@aulss5.veneto.it (A.M.)

**Keywords:** high–normal blood pressure, hypertension, lifestyle, ambulatory blood pressure, guidelines, nutraceuticals

## Abstract

(1) Background: Lifestyle changes, eventually coupled with a nutraceutical, are recommended strategies for managing high–normal blood pressure (BP) patients with low–moderate cardiovascular (CV) risk. In a real-life clinical setting, we evaluated the effects of generic written lifestyle advice, extrapolated from the 2018 ESC/ESH guidelines, and a beetroot-based nutraceutical on 24 h BP in a population with a high–normal office BP and low–moderate CV risk. (2) Methods: A longitudinal observational study was conducted in two ESH Hypertension Excellence Centres on 43 consecutive subjects with high–normal BP according to repeated office BP (OBP) measurements and a low–moderate CV risk based on SCORE2/SCORE2-OP. Additionally, 24 h ambulatory BP monitoring (ABPM) was carried out at baseline and three months after lifestyle changes, according to generic written advice from the 2018 ESC/ESH guidelines, coupled with a nutraceutical containing 500 mg of dry beetroot extract. (3) Results: The mean age was 50 ± 11 years, with male prevalence (54%). The prevalence of overweight/obesity was 58%. The mean OBP was 135 ± 3/85 ± 3 mmHg. At baseline, the mean 24 h BP, daytime BP, and night-time BP were 127 ± 7/80 ± 6 mmHg, 131 ± 8/83 ± 6 mmHg, and 118 ± 8/70 ± 5 mmHg, respectively, BP profiles compatible with hypertension status in some subjects. After a median follow-up of 98 (92–121) days, all BPs, except night-time diastolic BP, were significantly decreased: −3 ± 6/−2 ± 4 mmHg for 24 h BP, −3.9 ± 6.0/−3.0 ± 4.0 mmHg for daytime BP, and −3.3 ± 7.4/−1.3 ± 4.7 mmHg for night-time BP, respectively. No significant clinical changes in body weight were detected. BP decreased independently of baseline BP levels, sex, smoking status, and body mass index, while a more substantial BP decrease was observed in older patients. (4) Conclusions: Our exploratory study shows, for the first time, that written generic lifestyle advice taken from the ESC/ESH hypertension guidelines coupled with a beetroot-based nutraceutical may represent a valid initial non-pharmacological approach in subjects with a high–normal office BP and low–moderate CV risk, even without personalized diet interventions.

## 1. Introduction

The 2018 European Society of Cardiology (ESC)/European Society of Hypertension (ESH) Guidelines, as well as the latest 2023 ESH Guidelines for the management of arterial hypertension, support lifestyle changes as the initial therapeutic approach for individuals with high–normal blood pressure (BP) and grade 1 hypertension (systolic and diastolic BP (SBP, DBP) up to 159 and/or 99 mmHg, respectively) with low cardiovascular (CV) risk [1,2]. Lifestyle changes to lower BP values include reducing dietary calories, sodium and alcohol intake, adequate potassium integration, and physical activity, also aiming at reducing body weight and waist circumference in overweight/obese patients [3]. Although it has been proven that a healthy lifestyle can lower BP by approximately 4–5 mmHg [4], data about the effectiveness of strategies based on non-pharmacological interventions mainly come from investigations based on personalized diet and exercise programs by dieticians and trainers. In daily clinical practice, these large-scale approaches are strongly limited by high costs (many healthcare providers do not reimburse lifestyle strategies) and low adherence/persistence to the prescribed measures, as they may also interfere with home or working life habits and needs [5]. 

Alongside lifestyle changes, some dietary components, both as natural “functional” foods or as products in nutraceutical formulations, may have potential therapeutic properties in preventing or treating diseases [6] and are often complementarily administered, despite the low quality and strength of the supporting evidence (few or none RCTs, small sample sizes, short durations of follow-up, and surrogate biomarkers rather than patient outcomes) [7]. Nevertheless, a position document of the ESH stated that nutraceuticals may support lifestyle improvement in lowering BP without significant side effects [8]. Adherence to lifestyle interventions may be improved by the addition of a nutraceutical, especially in the setting of cardiovascular disease (CVD) prevention, where conventional pharmacological treatments and their possible adverse effects often assume a quod vitam connotation, even if cost-effectiveness may be detrimental in the long-term [9]. In arterial hypertension, nutritional supplements rich in inorganic nitrates (NO3-) have been found to exert antihypertensive properties. Among functional foods and nutraceuticals as sources of nitrates, beetroot juice and its by-products have received considerable attention, with presence in the literature of several studies, including placebo-controlled double-blinded RCTs and their metanalyses, exploring the effect of dietary nitrates beyond BP profile [10,11,12,13,14]. 

Based on these premises, we evaluated, in a real-life clinical study, the feasibility and effectiveness of a straightforward non-pharmacological approach based on generic written lifestyle advice extrapolated from the 2018 ESC/ESH guidelines, coupled with a beetroot-based nutraceutical, as a complementary part of dietary intervention, on the 24 h BP profile evaluated by ambulatory blood pressure monitoring (ABPM) in a population with high–normal office BP and low–moderate CV risk from two distinct ESH Hypertension Excellence Centres. 

## 2. Materials and Methods

### 2.1. Study Design and Population 

A bicentric, longitudinal, observational, open-label, and non-controlled exploratory study was conducted on consecutive patients referred to two Italian ESH “Hypertension Excellence Centres” (Internal Medicine and Geriatrics, IRCCS INRCA, Ancona, and Internal Medicine Clinical Unit, Rovigo General Hospital, Rovigo) from December 2021 to May 2022. The following inclusion criteria were applied: age ≥ 18 years, a high–normal office BP (SBP 130–139 mmHg and/or DBP 85–89 mmHg on repeated measures), as defined by the 2018 ESC/ESH guidelines [1], individuals with low–moderate CV risk, no history of previous CV events, no antihypertensive drug treatment, and willingness to adhere to the proposed lifestyle suggestions and a nutraceutical according to best clinical practice. We defined “low–moderate CV risk” as follows: patients aged <50 years with a calculated SCORE2 of <2.5%, patients aged 50–69 years with a calculated SCORE2 of <5%, and patients aged ≥70 years with a calculated SCORE2-OP of <7.5%, according to the 2021 ESC Guidelines on CVD prevention [15,16]. All participants gave their informed consent, and clinical investigations were conducted according to the principles expressed in the Declaration of Helsinki and its later amendments. This study was approved by the local institutional ethics committee (Comitato Etico INRCA; Approval Code: SC/14/443; Approval Date: 24 July 2014). 

### 2.2. Clinical Parameters 

The patients’ complete medical history, anthropometric measurements, laboratory parameters, and ABPM parameters were collected. Smoking habit was defined as current smoking or previous smoking of at least 100 cigarettes in a lifetime [17]. Body mass index (BMI) was defined as body mass divided by the square of body height and was expressed in units of kg per square meter. A BMI of <25 kg/m^2^ defined normal weight, while a BMI between 25 and 30 kg/m^2^ and a BMI of ≥30 kg/m^2^ defined overweight and obesity, respectively. After fasting blood sampling, the following laboratory parameters were taken into account: glucose, creatinine, and an estimation of glomerular filtration rate (eGFR) using the CKD-EPI equation, and lipid profile including total cholesterol (TC), high-density lipoprotein cholesterol (HDL-C), and triglycerides (TG). Low-density lipoprotein cholesterol (LDL-C) was calculated using the modified Friedewald equation proposed by Martin et al. [18].

### 2.3. Blood Pressure Measurements

We performed three sequential oscillometric automatic BP measurements on both arms during the office evaluation using validated devices (Microlife^®^ model BP3MQ1-2D and BP A200 AFib, Widnau, Switzerland). Correct cuff sizes (range 22–32 cm or 32–42 cm) were selected according to arm circumference, and BP measurements were performed after at least five minutes of rest in the sitting position. The patient’s arm was kept at the heart level during the measurement. The higher average BP value between arms was used for the analyses and to place the ABPM, thus avoiding errors due to interarm BP differences [19]. A 24 h ABPM was performed at baseline on each enrolled patient, using Spacelab devices model 90227 (Spacelab Healthcare, Snoqualmie, WA, USA) and TM-2430 ambulatory BP monitors (A&D Company, Tokyo, Japan) with appropriate cuff dimensions according to arm circumference. The mean 24 h BP, daytime BP (defined as the BP values from 06:00 to 22:00 h), and night-time BP (defined as the BP values from 22:00 to 06:00 h) were considered. The definitions of “day” and “night” periods were based on the most common answers to a questionnaire in which patients were asked about their sleeping behavior. Moreover, the medical staff verified the correct positioning of the brachial cuff and its proper functioning. The minimum quality criteria for a satisfactory ABPM recording were based on the recommendations made by Omboni et al. [20]. Patients with mean 24 h BP of <130/80 mmHg, mean daytime BP of <135/85 mmHg, and mean night-time BP of <120/70 mmHg were normotensive/controlled. We considered those patients with a mean SBP reduction equal to or greater than 10% from day to night as “dippers”.

### 2.4. Lifestyle Advice, Nutraceutical, and Follow-Up

At baseline, each enrolled subject received a summary with generic non-tailored dietary and lifestyle advice extrapolated from the 2018 ESC/ESH Guidelines on the management of arterial hypertension [1], written in simple, schematic Italian language for easy comprehension by any patient (see Appendix A online, which is a translation of the original form administered to the patients). The schematic summary included five items, each containing a brief text with an explanation and a figure. Such written advice was orally explained to the patients once, at the first visit. 

Furthermore, a once-daily (in the morning) nutraceutical containing red beetroot dry extract (500 mg of *Beta vulgaris* L.) was suggested as an integrated part of the lifestyle advice. Adding a nutraceutical to this lifestyle advice was intended as part of good clinical practice (GCP), as indicated by the recent ESH Position statement on nutraceuticals in hypertension [8]. No further treatments were proposed nor added during the observational period. After three months, a second 24 h ABPM was performed on each enrolled patient to re-evaluate the 24 h BP profile. The incidence of any adverse event was also evaluated at follow-up, and all enrolled patients were managed according to the guidelines and routine clinical practice; no other procedures or interventions were performed.

### 2.5. Statistical Analysis 

The data were analyzed using the Statistical Package for Social Science version 21 (SPSS Inc., Chicago, IL, USA). A *p*-value of less than 0.05 was defined as statistically significant. Continuous variables were checked for normality by double checking graphs, skewness, and kurtosis and were expressed as mean ± standard deviation (SD), or median and interquartile range (IQR) if markedly skewed. The χ^2^ test was used to analyze the differences between categorical variables at baseline. Student’s *T*-test and Mann–Whitney test were used to compare continuous variables at baseline. Paired *t*-test and McNemar test were used to assess the differences between the selected variables at the specified time intervals. To investigate the difference in ABP changes in the study population subgroups, we performed a repeated measures analysis of covariance (ANCOVA) using the ΔBP between the two times, adjusted for the basal BP values, to make the data independent from it. 

## 3. Results

### 3.1. Baseline Characteristics of the Study Population

We enrolled a cohort of 43 Caucasian patients with a mean age of 50 ± 11 years and a male prevalence (54%). The prevalence of overweight/obese patients was 58%. Twenty-two patients were enrolled in the Hypertension Centre of Ancona and twenty-one in Rovigo. Thus, ABPM identified 41.9% of the patients as having BP values compatible with hypertension, according to the 2018 ESC/ESH Guidelines [1], despite repeated measurements revealing only high–normal OBP before ABPM. As expected, hypertensive patients showed higher office and ambulatory blood pressure values, while no significant differences emerged in the other characteristics compared to normotensive patients, except for lower TC levels. The main clinical characteristics of the study population, the basal anthropometric measurements, and the laboratory parameters are reported in Table 1. 

### 3.2. Changes in ABPM at Follow-Up 

After a median follow-up of 98 (92–121) days, the overall BP profile improved, as shown by a statistically significant decrease in all ABPM parameters, except for mean night-time DBP (−3 ± 6/−2 ± 4 mmHg for 24 h BP; −4 ± 6/−3 ± 4 mmHg for daytime BP; and −3 ± 7/−1 ± 5 mmHg for night-time BP), as described in Figure 1. The prevalence of patients with a mean 24 h and mean daytime BP compatible with normotension increased to 75% and 86%, respectively. The population had non-clinically meaningful weight loss (∆weight = −0.4 ± 1.2 kg; *p* = 0.05) at follow-up. In the subgroup analyses, ABP decreased independently from sex, baseline BMI (<25 vs. ≥25 kg/m^2^), smoking status, and baseline BP (normotensives vs. hypertensives). Table 2 and Figure 2 show the ABPM changes at follow-up according to sex, baseline BP, BMI, and smoking status. Conversely, after dividing the study population according to the median age (median age: 51 years), a more significant reduction in 24 h SBP, daytime SBP, and daytime DBP was observed in the older subjects (aged >51 years) compared to the younger subjects (aged ≤51 years), as described in Figure 3. No significant difference in the prevalence of non-dipper subjects was found between baseline and follow-up (44.2% vs. 48.8%, *p* = 0.688). All the enrolled subjects completed the follow-up without any adverse events reported. 

## 4. Discussion

In this open-label, single-group, exploratory, longitudinal study, we explored, in a real-life clinical scenario, the feasibility and effectiveness of lifestyle advice promoted by the 2018 ESC/ESH Guidelines in a sample of subjects with high–normal office BP evaluated with ABPM, the most reproducible and reliable method for assessing BP in clinical practice [1]. 

Evidence has shown how lifestyle modifications are safe and effective, with benefits beyond CV health. Regarding CVD, they are recommended as first-line preventive and treatment strategies for managing hypertension, dyslipidemias, and type 2 diabetes mellitus (DM2), regardless of their severity and the individual basal CV risk [15]. In some cases, mainly through weight and visceral adiposity reduction, a healthy lifestyle can completely reverse the dysmetabolic and neuro-hormonal derangement that characterizes CV risk factors, avoiding further progression and reducing the need for pharmacological treatments. In the specific setting of arterial hypertension, adequate lifestyle interventions should be suggested in high–normal BP patients, who represent nearly 30% of the general population [21]. Intercepting patients at these early stages is pivotal because it has been demonstrated that overt hypertension may develop in up to 65% of non-treated high–normal BP cases within the following two to four years [20] and because high–normal BP has been associated with hypertension-related CV risk and target organ damage, similar to that typically found in frankly hypertensive individuals [22,23]. Indeed, as emerged from the ABPM performed in our high–normal OBP population, most of the patients already showed ABP profiles compatible with hypertension, unveiling masked hypertension. The 2018 ESC/ESH Guidelines for managing arterial hypertension proposed general lifestyle recommendations, including suggestions on preferable food groups, limited salt assumption, moderate alcohol consumption, quitting smoking, and performing regular physical activity [2]. All these recommendations have been adopted in our study, showing how even generic written and explained advice can potentially lower BP values. 

Although each single lifestyle intervention is recommended by the ESC/ESH Guidelines in class 1 with an A or B level of evidence throughout the spectrum of BP severity, starting from high–normal BP or grade 1 hypertension [1], evidence of the effectiveness of their summary in simple generic advice in daily clinical practice is still scarce, if not absent [24]. Our study attempted to evaluate this issue. Indeed, most investigations reporting positive effects of lifestyle changes on BP have usually been conducted using tailored approaches based on personalized diet and exercise program interventions according to individual characteristics and needs. An example is the vast literature on the Mediterranean diet and BP [25]. In this setting, more successful interventions emerged when personalization included direct counselling with dieticians and trainers, scheduled interventions on food and exercise, and a stricter follow-up. Conversely, our study showed how a more straightforward and low-cost approach based on written generic advice extrapolated from the 2018 ESC/ESH Guidelines on arterial hypertension and endorsed by the most recent 2023 ESH Guidelines could reduce BP in patients with high–normal OBP and low–moderate CV risk. An overall significant reduction in ABP was found after a 3-month follow-up.

Our study used a nutraceutical as a complementary part of the lifestyle advice, according to the same guidelines and the ESH position paper for this type of patient [8]. Previous observational investigations, RCTs, and their meta-analyses have evaluated beetroot-based nutraceuticals’ efficacy and safety in lowering BP in low-CV-risk patients, with encouraging data [12,26,27]. In particular, beetroot consumption in the form of juice seemed to be associated with significant dose-dependent changes in SBP (mean reduction −4 mmHg, 95% CI −6 to −3) in a meta-analysis of RCTs with variable duration (2 h–15 days) and different daily doses ranging from 321 to 2790 mg [10]. Other investigations were also consistent with a significant DBP reduction, probably related to increased plasma nitrate and nitrite concentrations [14,28]. As mentioned, BP changes in response to beetroot or other dietary sources of nitrates administration evaluated with ABPM led to appreciably variable results [13,14]. Our study, without a placebo control group instead of the nutraceutical, cannot discern the contribution given to the BP lowering by the lifestyle advice from that of the nutraceutical itself. However, evaluating the effectiveness of the nutraceutical, as an active substance effective in reducing BP, was not within the objectives of our study. In our exploratory study, the adherence to the nutraceutical was high, given that all patients declared to have taken it (85% of patients reported full adherence in the entire study period).

It is important to note that our original and clinically non-negligible data are preliminary and require further confirmation, not allowing us to give definitive conclusions. The placebo effect is a benefit experienced by the patient taking an inert substance due to the expectation of benefit [29]. Focusing our attention on lifestyle interventions based on generic advice, the concept of a placebo control group is not easily applicable in our setting, in which no substance or set of substances or precise and standardized interventions are proposed. Conversely, it would also be unethical to suggest to these patients a placebo/usual diet, which is classically rich in sodium and low in potassium [30]. At the same time, there is extensive evidence of the benefits of diets that are low in sodium and rich in potassium, as indicated in the 2018 ESC/ESH Guidelines and our written advice. 

In our study, a significant decrease in body weight did not emerge at follow-up. Therefore, the reason for the BP reduction should be sought elsewhere. Indeed, if implemented, the suggestion to limit overall salt intake, mainly from pre-salted foods, along with the advice to eat foods rich in potassium and perform physical activity, might well explain the BP lowering [31,32]. Very recently, a crossover study showed how low- vs. high-sodium diets (approximately 500 mg daily total vs. around 2200 mg sodium added daily to usual diet) led to a median within-individual reduction in mean BP of about four mmHg, independently from hypertension status and antihypertensive medication use [31]. Following the 3-month visit, we asked patients for feedback on the suggestions given. Since the study was based on generic advice and did not require a diary from patients regarding their lifestyle (which is not usually performed in daily clinical practice), the data obtained from the feedback were based on the simple subjective qualitative responses of the patients and their reliability. Most patients (31/43) reported adhering to at least one of the lifestyle recommendations provided in the follow-up period, with more than half of the patients declaring that they had increased physical activity (26/43 patients), adapted to a healthy diet (26/43 patients), and/or reduced alcohol consumption (25/43 patients). Only 4 out of 14 smokers declared that they had stopped smoking or, in any case, reduced the number of cigarettes smoked per day. Most patients confirmed their willingness to adhere to the simple written lifestyle advice after the observational period.

To the best of our knowledge, the presence of a placebo effect regarding dietary interventions has not been studied in the literature. The evidence of its existence is almost absent and at a high risk of bias [33,34]. However, in our study, we cannot exclude that some possible placebo effect may have contributed to our findings, although this does not change the final message: the generic and non-tailored advice cited in many guidelines (in our case, the 2018 ESC/ESH for the management of hypertension) reported on the written form given and explained to the patients, strengthened by a nutraceutical (in this case a beetroot-based one), can be helpful for lowering BP in clinical practice in a carefully selected population (in this case, patients with high–normal office BP and low–moderate CV risk). 

Regarding the subgroup analyses, patients with initial ABPM profiles compatible with hypertension and non-smokers showed a greater but non-statistically significant BP decrease. At the same time, older individuals had a more considerable BP reduction. The study design only allows us to make limited hypotheses about these findings. Smoking and ageing partly share the mechanisms sustaining hypertension, as both lead to a loss of elasticity, increased arterial stiffness, and reduced endothelium-dependent vasodilation. The known “regression to the mean” effect can also have a role. Alongside different adherence to lifestyle advice, different arterial responses to vasodilating stimuli by restoring endothelial function and enhancing NO bioavailability could have brought out this additional finding, as reported by recent experimental and clinical investigations [35,36,37]. Thus, middle-aged pre-hypertensive and non-smoking patients could be the preferable target of this approach based on generic lifestyle advice and nutraceutical supplementation [38]. 

Regarding the safety aspect, no adverse events were noted, as expected. Enhanced adherence aligns with a recent study considering and comparing lipid-lowering therapies (LLTs), which revealed 30% more persistence at two years of non-reimbursed nutraceuticals versus statins, independently of age, adverse events during treatment, and estimated CV risk [39]. In our real-life study, we did not intentionally use a precise tool to evaluate adherence to both lifestyle advice (e.g., 24 h urinary sodium excretion) and nutraceutical (e.g., rating scales), because they would have affected the patients’ behaviors during follow-up, while our study primarily aimed to investigate the effectiveness of written generic lifestyle advice in daily clinical practice. 

Based on our positive and original findings, future studies could explore this approach, perhaps by comparing it with hypothetical control groups composed of patients with a different lifestyle intervention (i.e., patients with personalized diet programs or written advice based on other guidelines). 

### Study Strengths and Limitations

The original but straightforward design of the study based on the ESC/ESH guidelines and the use of ABPM to evaluate BP profiles at baseline and follow-up, allowing for a greater accuracy in assessing BP changes and control [40], along with being bicentric, represent the main strengths of our study. Given the small sample size and the absence of a control group that defines the design as exploratory, our findings should be considered as preliminary and interpreted accordingly. However, despite the small sample taken into account, based on the change in 24 h systolic BP at follow-up found and the significance level set, our study has a statistical power of 90%. Nevertheless, the encouraging results found in our study may represent a base for further investigations. We also hope our guidelines-based approach can be easily applied and reproduced to benefit this common type of patient.

## 5. Conclusions

Our study showed that written generic lifestyle interventions extrapolated from the 2018 ESC/ESH Guidelines coupled with a beetroot-based nutraceutical led to a mid-term ABP profile normalization in individuals with a high–normal BP, reducing progression to overt “office” hypertension and the need for pharmacological treatment. To our knowledge, no previous studies have evaluated the effectiveness of simple, written, generic, non-personalized lifestyle advice in daily clinical practice. In contrast, most studies on hypertensives are instead focused on specific, standardized, or personalized diets. Despite the limitations mentioned above, our findings support that generic lifestyle advice associated with a nutraceutical source of nitric oxide may be a valid initial non-pharmacological approach in subjects with a high–normal BP and low–moderate CV risk, many of which proved to be grade 1 hypertensives at ABPM. These results align with the growing evidence that lifestyle interventions are effective and safe and should be intended broadly, with some nutraceuticals playing a complementary role in enhancing the effectiveness of more conventional approaches and helping to overcome limitations such as poor adherence. 

## Figures and Tables

**Figure 1 nutrients-15-05099-f001:**
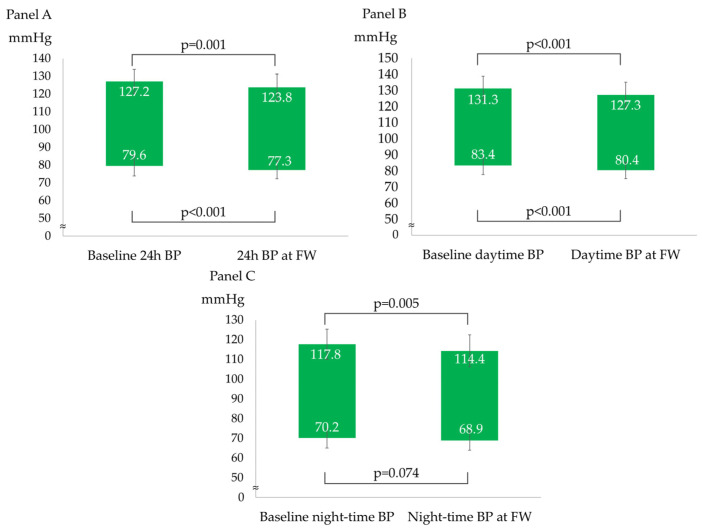
ABPM parameters changes from baseline to follow-up. (**A**) 24 h systolic and diastolic blood pressure changes; (**B**) daytime systolic and diastolic blood pressure changes; and (**C**) night-time systolic and diastolic blood pressure changes.

**Figure 2 nutrients-15-05099-f002:**
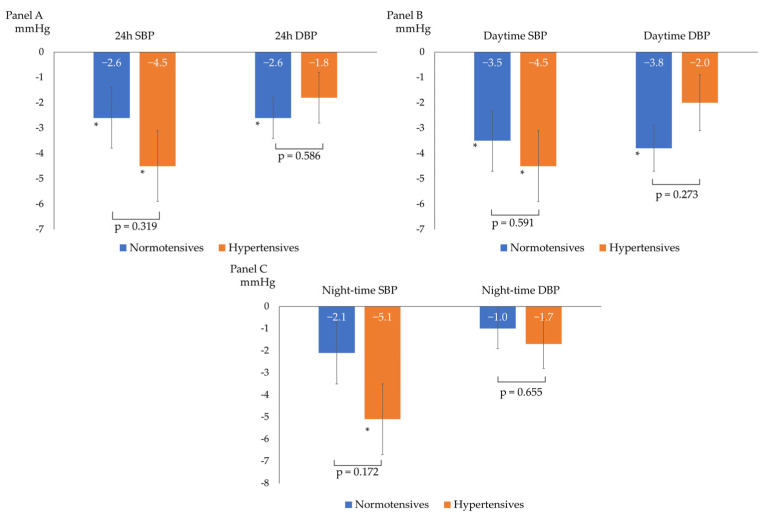
Changes from baseline ABP according to baseline blood pressure (normotensives vs. hypertensives). (**A**) 24 h systolic (SBP) and diastolic (DBP) blood pressure changes; (**B**) daytime systolic (SBP) and diastolic (DBP) blood pressure changes; and (**C**) night-time systolic (SBP) and diastolic (DBP) blood pressure changes; * *p* < 0.05 for comparison between baseline and follow-up BP within the single subgroup. The *p*-values shown refer to the *p* for interaction.

**Figure 3 nutrients-15-05099-f003:**
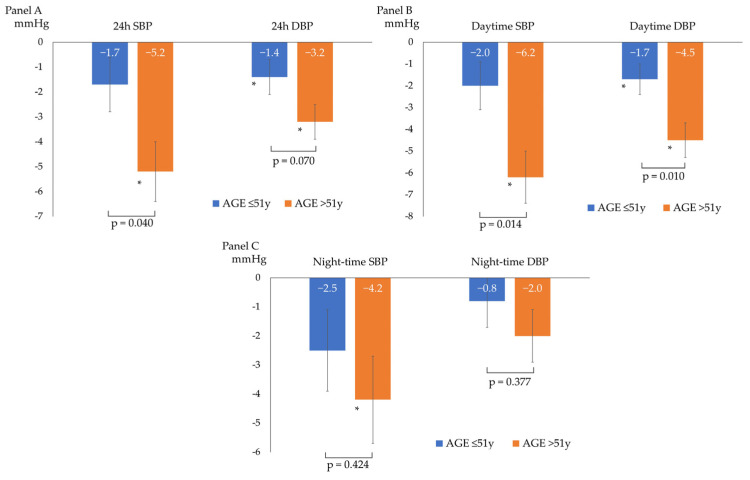
Changes from baseline ABP according to the median age of the study population: ≤ 51 years (*n* = 23 patients) or > 51 years (*n* = 20 patients). (**A**) 24 h systolic (SBP) and diastolic (DBP) blood pressure changes; (**B**) daytime systolic (SBP) and diastolic (DBP) blood pressure changes; (**C**) and night-time systolic (SBP) and diastolic (DBP) blood pressure changes; * *p* < 0.05 for comparison between baseline and follow-up BP within the single subgroup. The *p*-values shown refer to the *p* for interaction.

**Table 1 nutrients-15-05099-t001:** Baseline clinical and laboratory parameters of the entire study population and according to hypertension status.

	All Population (n° 43)	Normotensives at ABPM ^1^ (n° 25)	Hypertensives at ABPM ^1^ (n° 18)	*p* (Normotensives vs. Hypertensives)
Age (years)	49.9 ± 10.8	50.4 ± 10.7	49.2 ± 11.0	0.726
Sex (male)	23 (53.5%)	13 (52.0%)	10 (55.6%)	0.818
BMI ^2^ (kg/m^2^)	25.1 ± 3.2	25.0 ± 3.0	25.4 ± 43.5	0.708
Weight (kg)	73.0 ± 11.4	72.6 ± 11.2	73.5 ± 11.9	0.802
Office SBP ^3^ (mmHg)	135.4 ± 3.9	135.3 ± 2.6	135.6 ± 3.5	0.829
Office DBP ^4^ (mmHg)	85.1 ± 3.1	84.7 ± 2.9	85.9 ± 3.7	0.356
24 h SBP ^3^ (mmHg)	127.2 ± 6.8	124.8 ± 5.6	130.6 ± 7.0	0.005
24 h DBP ^4^ (mmHg)	79.6 ± 5.6	76.2 ± 3.3	84.3 ± 4.5	<0.001
Daytime SBP ^3^ (mmHg)	131.3 ± 7.5	128.9 ± 6.7	134.5 ± 7.5	0.015
Daytime DBP ^4^ (mmHg)	83.4 ± 5.5	80.1 ± 3.3	88.0 ± 4.7	<0.001
Night-time SBP ^3^ (mmHg)	117.8 ± 7.7	116.1 ± 6.2	120.1 ± 9.1	0.091
Night-time DBP ^4^ (mmHg)	70.2 ± 5.0	68.0 ± 2.9	73.2 ± 5.6	0.002
Glucose (mmol/L)	5.11 ± 0.50	5.24 ± 0.46	4.93 ± 0.53	0.104
Creatinine (µmol/L)	77.4 ± 16.7	73.0 ± 14.1	83.6 ± 18.5	0.063
eGFR ^5^ (mL/min/1.73 m^2^)	94.9 ± 13.0	97.6 ± 11.6	90.9 ± 14.3	0.139
TC ^6^ (mmol/L)	5.15 ± 0.39	5.30 ± 0.35	4.92 ± 0.33	0.004
HDL-C ^7^ (mmol/L)	1.53 ± 0.31	1.54 ± 0.37	1.52 ± 0.26	0.856
LDL-C ^8^ (mmol/L)	3.13 ± 0.47	3.24 ± 0.48	2.96 ± 0.39	0.088
TG ^9^ (mmol/L)	0.99 (0.84–1.21)	1.08 (0.90–1.35)	0.92 (0.77–1.08)	0.111

All the continuous variables are expressed as mean ± SD, except for TG, which is expressed as median and interquartile range (IQR), given its non-normal distribution. ^1^ Ambulatory Blood Pressure Monitoring, ^2^ Body Mass Index, ^3^ Systolic Blood Pressure, ^4^ Diastolic Blood Pressure, ^5^ estimated Glomerular Filtration Rate, ^6^ Total Cholesterol, ^7^ High-Density Lipoprotein Cholesterol, ^8^ Low-Density Lipoprotein Cholesterol, ^9^ Triglycerides.

**Table 2 nutrients-15-05099-t002:** Changes from baseline ABP according to sex, BMI, and smoking status.

ABPM ^1^ (mmHg)	Sex		Baseline BMI ^2^ (kg/m^2^)		Smoking Status	
	Male (n° 23)	Female (n° 20)	<25 (n° 18)	≥25 (n° 25)	No (n° 29)	Yes (n° 14)
24 h SBP ^3^	−3.5 ± 1.2 *	−3.2 ± 1.3 *	−3.9 ± 1.3 *	−3.0 ± 1.1 *	−4.0 ± 1.0 *	−2.1 ± 1.5
	*p* for interaction 0.873		*p* for interaction 0.635		*p* for interaction 0.311	
24 h DBP ^4^	−2.2 ± 0.7 *	−2.2 ± 0.7 *	−2.2 ± 0.8 *	−2.2 ± 0.7 *	−2.6 ± 0.6 *	−1.5 ± 0.9
	*p* for interaction 0.991		*p* for interaction 0.997		*p* for interaction 0.312	
Daytime SBP	−4.1 ± 1.2 *	−3.8 ± 1.3 *	−4.2 ± 1.4 *	−3.7 ± 1.2 *	−4.7 ± 1.0 *	−2.4 ± 1.5
	*p* for interaction 0.862		*p* for interaction 0.798		*p* for interaction 0.229	
Daytime DBP	−3.0 ± 0.8 *	−3.1 ± 0.8 *	−3.1 ± 0.9 *	−3.0 ± 0.7 *	−3.6 ± 0.7 *	−1.9 ± 1.0
	*p* for interaction 0.909		*p* for interaction 0.900		*p* for interaction 0.151	
Night-time SBP	−3.4 ± 1.4 *	−3.3 ± 1.5 *	−4.0 ± 1.6 *	−2.8 ± 1.4 *	−3.9 ± 1.3 *	−2.2 ± 1.8
	*p* for interaction 0.965		*p* for interaction 0.595		*p* for interaction 0.447	
Night-time DBP	−1.1 ± 0.9	−1.5 ± 1.0	−1.3 ± 1.0	−1.3 ± 0.8	−1.4 ± 0.8	−1.1 ± 1.1
	*p* for interaction 0.772		*p* for interaction 0.981		*p* for interaction 0.808	

Repeated measures analysis of covariance (ANCOVA) was adjusted for the basal BP values. * *p* < 0.05 for comparison between baseline and follow-up BP within the single subgroup. ^1^ Ambulatory Blood Pressure Monitoring, ^2^ Body Mass Index, ^3^ Systolic Blood Pressure, ^4^ Diastolic Blood Pressure.

## Data Availability

The datasets used and analyzed during the current study are available from the corresponding author upon reasonable request.

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
