# Peer review of "The Effects of ESC/ESH-Based Written Generic Lifestyle Advice and a Nutraceutical on 24-Hour Blood Pressure in Patients with High–Normal Office Blood Pressure and Low–Moderate Cardiovascular Risk"

_nutrients, 2023, doi:10.3390/nu15245099_

Round 1

Reviewer 1 Report

Comments and Suggestions for Authors

Comments provided in round one of the submission were adequately addressed. Thanks to the authors for their attention to these comments.

Author Response

We thank your precious collaboration.

Reviewer 2 Report

Comments and Suggestions for Authors

1) Statistical analysis - paired t-test was used, it is puzzling that the clinical data had a normal distribution, did the author make any data transformation (eg. logarithmisation)?
2) Table 1 - values in SI units should be provided, TG is given in IQR, but other parameters is given as a difference between third and first quartile - please unify 
3) Line 191-202 is completely unreadable and hard to follow - why authors don't present p values on graphs or on separate table 
4) Results - how many patients were >52 years? why this value is used as a cut-off ? median is 50 (table 1)
5) I could not find information whether changing of lifestyle was verified during control visit after 3 months? Have patients followed the recommendations?

Author Response

1) Statistical analysis - paired t-test was used, it is puzzling that the clinical data had a normal distribution, did the author make any data transformation (eg. logarithmisation)?

Answers/comments:

Blood pressure (BP) generally assumes a normal distribution in the population, and this has also been confirmed in the present study, where the normal distribution of the BP variables has been verified through double-checking using graphs, skewness and kurtosis.

To investigate the ambulatory BP trends in the subgroups of our study population, we performed the repeated measures analysis of covariance (ANCOVA) using the "delta" () BP between the two time periods adjusted for the basal BP values. In this context, ANCOVA is the best statistical test (no non-parametric statistical analyses can evaluate the presence of differences in the trend of a continuous variable over time between different groups). Although not all BP variables are within the values considered optimal/excellent for skewness and kurtosis (+/-1), their distributions remain within skewness and kurtosis values deemed acceptable to prove normal distribution. Moreover, after trying to refine further the normalization of the BP variables using logarithm, square root or other mathematical tools, their distributions were significantly more skewed.

Several authors consider acceptable values for skewness and kurtosis between -2 and +2 to prove normal distribution [George, D. & Mallery, M. (2010). SPSS for Windows Step by Step: A Simple Guide and Reference, 17.0 update (10a ed.) Boston: Pearson], while others argue that distribution of data can be considered normal if skewness is between ‐2 to +2 and kurtosis is between ‐7 to +7 [Hair, J., Black, W. C., Babin, B. J. & Anderson, R. E. (2010) Multivariate data analysis (7th ed.). Upper Saddle River, New Jersey: Pearson Educational International. Byrne, B. M. (2010). Structural equation modelling with AMOS: Basic concepts, applications, and programming. New York: Routledge]. Furthermore, other authors suggested that the absolute value of skewness and kurtosis should not be greater than 3 and 10, respectively [Kline, R. B. (2011). Principles and Practice of Structural Equation Modeling (5th ed., pp. 3–427). New York: The Guilford Press].

Keeping these considerations in mind, all the BP distributions of our study (baseline and follow-up) showed a skewness between -1 and +1 and a kurtosis between -1 and +1, except for 24h-SBP and daytime-SBP, in whom kurtosis was 2.447 and 2.191, respectively. Regarding the BP variables, the skewness was between -1 and +1, except for night-time SBP (-1.735) and 24hSBP (-1.329), while the average kurtosis was around |3|, with the only exception of the night-time SBP kurtosis approaching the limit of +7 (6.237).  

In the revised manuscript, we have better specified the method used for normality check in the statistical analysis section.  

2) Table 1 - values in SI units should be provided, TG is given in IQR, but other parameters is given as a difference between third and first quartile - please unify

Answers/comments:

We are sorry for the misunderstanding. In Table 1, the continuous variables with a normal distribution have been expressed as mean ± SD. In contrast, the non-normally distributed variables (such as triglycerides) have been described as the median and interquartile range (IQR), as specified in the statistical analysis section.

In the revised Table 1, we have better clarified this aspect in the footnotes and provided the SI units, as suggested.

3) Line 191-202 is completely unreadable and hard to follow - why authors don't present p values on graphs or on separate table

Answers/comments:

Thank you for the suggestion. In the mentioned lines, we presented the p-values for the interaction of the ANCOVA on the different subgroups. We agree with you that the sentence may not be easily readable; therefore, in the revised manuscript, we moved this statistical information to the relative table (Table 2) and figures (see both "new" Figure 2 and "new" Figure 3).

4) Results - how many patients were >52 years? why this value is used as a cut-off ? median is 50 (table 1)

Answers/comments:

We apologize for the misunderstanding. The mean age of the study population was 50 years, as reported in Table 1, while the median age was 51 years. The study population was then divided based on the median age using the appropriate function of the SPSS software. The analysis in Figure 2 (now "new" Figure 3) was performed by comparing patients who were 51 years old (n = 23) with those who were > 51 years old (n = 20), based on the median age of the population. We better specified this aspect in the "new" Figure 3 and in the text of the revised manuscript.

5) I could not find information whether changing of lifestyle was verified during control visit after 3 months? Have patients followed the recommendations?

Answers/comments:

We aimed to evaluate, in a real-life clinical study (and not in a randomized controlled double-blinded study), the feasibility and effectiveness of a non-pharmacological approach based on generic written lifestyle indications taken from the 2018 ESC/ESH Guidelines in patients with high-normal BP and low cardiovascular risk, differently from what has been usually done in most previous investigations conducted using tailored approaches based on personalized diets and exercise program interventions. Thus, the present study's primary focus was a "non-structured" or "minimally structured" approach to lifestyle advice, formally based on the European Guidelines suggestions. Indeed, after the oral explanation and the printed summary given to the subjects, we deliberately left the patients to their free will.

We intentionally did not use a precise tool to evaluate the adherence to our suggestions because it would have affected the patients' behaviours during follow-up, while our study primarily aimed to investigate the effectiveness of written generic lifestyle advice in daily clinical practice. For example, an excellent tool to verify the change to a correct diet (low in sodium and rich in potassium) could have been the measurement of 24-hour urinary sodium and potassium levels. However, the patient's awareness of having to perform this lab examination could have affected their approach to the generic lifestyle indications, thus not reflecting the real-life clinical practice, where the routine testing of 24-hour urinary sodium and potassium excretion is not performed. We certainly know that a significant decrease in body weight did not emerge at follow-up in our study. Therefore, the reason for the BP reduction should be sought elsewhere (probably the limited overall salt intake, mainly from the decrease in pre-salted foods consumption, along with the suggestion to eat foods rich in potassium and to perform physical activity).

All the enrolled subjects completed the follow-up period and confirmed their willingness to adhere to the simple written lifestyle advice after the observational period. Following the 3-month visit, we asked patients for feedback on the suggestions given. Since the study was based on generic advice and did not require a diary from patients regarding their lifestyle (which is not usually done in clinical practice), the data obtained from the feedback were based on the simple subjective qualitative responses of the patients and their reliability. Therefore, a statistical analysis was not carried out on these data, given the lack of objective data and the sample size did not allow it. At the end of the study, most subjects declared good adherence to the proposed anti-hypertensive non-pharmacological strategy, as reported in the text. In the revised manuscript, we have better detailed the feedback of the patients (see discussion section) and all these aspects.

Reviewer 3 Report

Comments and Suggestions for Authors

In the present manuscript, authors explore the effect of written lifestyle advice on blood pressure (BP) reduction in individuals with high normal office BP and low-moderate CV risk.

Although it is a small, observational study with several limitations, the conclusions merit attention.

It is well-written study, however there are issues that should be addressed:

1. Title: Effects of....in patients with high-normal office blood pressure (office BP is more accurate given that almost 50% of those patients were categorized as hypertensives (stage 1) according to ABPM).

2. Authors did not measure adherence to the lifestyle recommendations. Although this may not be feasible for some parameters, such as the amount of salt consumption, information on adaptation of a healthy diet (eg. information on the frequency of fruit or vegetable consumption) and lifestyle behaviors (eg. physical activity) would be desirable. Moreover, are there data on smoking cessation? How did you confirm that the nutraceutical beetroot dry extract was taken by the participants?

3. Replace the term 'glycaemia' with 'glucose'. In Table 1, add the value of office mean BP and ABPM results for 24h, day-time and night-time BP (remove this information from the Results). In the first paragraph of the Results, presentation of the difference in baseline characteristics of patients with Hypertension stage 1 (according to ABPM) and normotensives would be advisable.

4. Figure 1, Figure 3. ABPM parameters change (not variations). In Figure 3, add the change in BP parameters according to baseline BP (hypertensives vs normotensives).

5. In Table 2, it would be more appropriate to present the difference in BP values in the total population, before and after lifestyle recommendations, (BP before, BP after and p-value) adjusted for age,gender, smoking, BMI and baseline BP values.

6. Did you calculate sample size?

Comments on the Quality of English Language

This is a well-written study.

Author Response

In the present manuscript, authors explore the effect of written lifestyle advice on blood pressure (BP) reduction in individuals with high normal office BP and low-moderate CV risk.

Although it is a small, observational study with several limitations, the conclusions merit attention.

It is well-written study, however there are issues that should be addressed:

  1. Title: Effects of....in patients with high-normal office blood pressure (office BP is more accurate given that almost 50% of those patients were categorized as hypertensives (stage 1) according to ABPM).

Answers/comments:

Thank you for the comment. We amended the title of the paper, as you suggested, which most closely reflects the BP characteristics of the study population.

  1. Authors did not measure adherence to the lifestyle recommendations. Although this may not be feasible for some parameters, such as the amount of salt consumption, information on adaptation of a healthy diet (eg. information on the frequency of fruit or vegetable consumption) and lifestyle behaviors (eg. physical activity) would be desirable. Moreover, are there data on smoking cessation? How did you confirm that the nutraceutical beetroot dry extract was taken by the participants?

Answers/comments:

Thank you for the comment. As also replied to Reviewer 2, we aimed to evaluate, in a real-life clinical study (and not in a randomized controlled double-blinded study), the feasibility and effectiveness of a non-pharmacological approach based on generic written lifestyle indications taken from the 2018 ESC/ESH Guidelines in patients with high-normal BP and low cardiovascular risk, differently from what has been usually done in most previous investigations conducted using tailored approaches based on personalized diets and exercise program interventions. Thus, the present study's primary focus was a "non-structured" or "minimally structured" approach to lifestyle advice, formally based on the European Guidelines suggestions. Indeed, after the oral explanation and the printed summary given to the subjects, we deliberately left the patients to their free will.

We intentionally did not use a precise tool to evaluate the adherence to our suggestions because it would have affected the patients' behaviours during follow-up, while our study primarily aimed to investigate the effectiveness of written generic lifestyle advice in daily clinical practice. For example, an excellent tool to verify the change to a correct diet (low in sodium and rich in potassium) could have been the measurement of 24-hour urinary sodium and potassium levels. However, the patient's awareness of having to perform this lab examination could have affected their approach to the generic lifestyle indications, thus not reflecting the real-life clinical practice, where the routine testing of 24-hour urinary sodium and potassium excretion is not performed. We certainly know that a significant decrease in body weight did not emerge at follow-up in our study. Therefore, the reason for the BP reduction should be sought elsewhere (probably the limited overall salt intake, mainly from the decrease in pre-salted foods consumption, along with the suggestion to eat foods rich in potassium and to perform physical activity).

All the enrolled subjects completed the follow-up period and confirmed their willingness to adhere to the simple written lifestyle advice after the observational period. Following the 3-month visit, we asked patients for feedback on the suggestions given. Since the study was based on generic advice and did not require a diary from patients regarding their lifestyle (which is not usually done in clinical practice), the data obtained from the feedback were based on the simple subjective qualitative responses of the patients and their reliability. Therefore, a statistical analysis was not carried out on these data, given the lack of objective data and the sample size did not allow it. At the end of the study, most subjects declared good adherence to the proposed anti-hypertensive non-pharmacological strategy, as reported in the text. In the revised manuscript, we have better detailed the feedback of the patients (see discussion section) and all these aspects.

  1. Replace the term 'glycaemia' with 'glucose'. In Table 1, add the value of office mean BP and ABPM results for 24h, day-time and night-time BP (remove this information from the Results). In the first paragraph of the Results, presentation of the difference in baseline characteristics of patients with Hypertension stage 1 (according to ABPM) and normotensives would be advisable.

Answers/comments:

We replaced the term 'glycaemia' with 'glucose' in the revised manuscript. We added the values of office and ambulatory BPs in Table 1. Moreover, we reported the difference in baseline characteristics of patients with Hypertension stage 1 (according to ABPM) and normotensives in Table 1, describing them in the text, to maintain a good readability of the results section.

  1. Figure 1, Figure 3. ABPM parameters change (not variations). In Figure 3, add the change in BP parameters according to baseline BP (hypertensives vs normotensives).

Answers/comments:

In the revised Figure 1, we replaced the term "variations" with "change", as suggested. In the previous version, the change in BP parameters according to baseline BP (hypertensives vs normotensives) was described in Table 2, together with the other subgroup analyses with non-significant differences (see p for interaction in the "old" Table 2). The only subgroup analysis we chose to show as a Figure was the one according to age since it was the only statistically significant. In the revised manuscript, we moved the ANCOVA according to baseline BP from Table 2 to the "new" Figure 2 (the "old" Figure 2 has now become the "new" Figure 3), as suggested. 

  1. In Table 2, it would be more appropriate to present the difference in BP values in the total population, before and after lifestyle recommendations, (BP before, BP after and p-value) adjusted for age, gender, smoking, BMI and baseline BP values.

Answers/comments:

The analysis of the general population has been already described in Figure 1. The purpose of Table 2 and "new" Figures 2-3 is to evaluate whether the BP decrease was uniform among the various subgroups or whether there were differences. Table 2 has already been amended following the suggestions made by the Reviewer 2.

The sample size does not allow us to perform a repeated measures analysis of covariance (ANCOVA) with all the adjustments you requested in the same model. Furthermore, in our opinion, our subgroup analysis is more informative for our purpose and more appropriate. To evaluate the amount of BP change from baseline in the subgroups, we considered the "delta" () BP, adjusted for the baseline BP value. The BP change has been adjusted for the baseline BP values, which could differ between the two comparison groups and affect the results. The same evaluation could not be done by comparing the absolute values of BP before and after lifestyle recommendations in the subgroups.

  1. Did you calculate sample size?

Answers/comments:

Thank you for the question. As reported in the manuscript, our work is configured as an exploratory study. No previous studies have evaluated the effectiveness of simple written, generic, non-personalized lifestyle advice in daily clinical practice. In contrast, most studies on hypertensives focused on specific, standardized or personalized approaches.

Our study was conducted on 43 patients. Considering the change in 24-hour systolic BP at follow-up (from 127 ± 6.7 mmHg to 123 ± 7.3 mmHg) as the principal analysis, a significance level set at 0.05 and this sample size, our study has a 90% statistical power (calculated by G*power statistical software). We reported this aspect in the revised manuscript.